# Event Traffic Forecasting with Sparse Multimodal Data

Xiao Han
University of Chinese Academy of
Sciences
Beijing, China
Pengcheng Laboratory
Shenzhen, China
hanxiao22@mails.ucas.ac.cn

Zhenduo Zhang
University of Chinese Academy of
Sciences
Beijing, China
Pengcheng Laboratory
Shenzhen, China
zhangzhenduo21@mails.ucas.ac.cn

Yiling Wu*
Pengcheng Laboratory
Shenzhen, China
wuyl02@pcl.ac.cn

Xinfeng Zhang*
University of Chinese Academy of
Sciences
Beijing, China
xfzhang@ucas.ac.cn

Zhe Wu
Pengcheng Laboratory
Shenzhen, China
wuzh02@pcl.ac.cn

## Abstract

With the development of deep learning, traffic forecasting technology has made significant progress and is being applied in many practical scenarios. However, various events held in cities, such as sporting events, exhibitions, concerts, etc., have a significant impact on traffic patterns of surrounding areas, causing current advanced prediction models to fail in this case. In this paper, to broaden the applicable scenarios of traffic forecasting, we focus on modeling the impact of events on traffic patterns and propose an event traffic forecasting problem with multimodal inputs. We outline the main challenges of this problem: diversity and sparsity of events, as well as insufficient data. To address these issues, we first use textual modal data containing rich semantics to describe the diverse characteristics of events. Then, we propose a simple yet effective multi-modal event traffic forecasting model that uses pre-trained text and traffic encoders to extract the embeddings and fuses the two embeddings for prediction. Encoders pre-trained on large-scale data have powerful generalization abilities to cope with the challenge of sparse data. Next, we design an efficient large language model-based event description text generation pipeline to build multi-modal event traffic forecasting datasets, ShenzhenCEC and SuzhouIEC. Experiments on two real-world datasets show that our method achieves state-of-the-art performance compared with eight baselines, reducing mean absolute error during the event peak period by 4.26%. Code is available at: https://github.com/2448845600/EventTrafficForecasting.

## CCS Concepts

• **Information systems** → **Spatial-temporal systems**; *Data mining*; • **Computing methodologies** → **Artificial intelligence**.

---

*Corresponding author

*MM '24, October 28-November 1, 2024, Melbourne, VIC, Australia*
© 2024 Copyright held by the owner/author(s).
ACM ISBN 979-8-4007-0686-8/24/10
https://doi.org/10.1145/3664647.3680706

## Keywords

Dataset, Multimodal Fusion, Time-Series and Language, Event Traffic Forecasting

**ACM Reference Format:**
Xiao Han, Zhenduo Zhang, Yiling Wu, Xinfeng Zhang, and Zhe Wu. 2024. Event Traffic Forecasting with Sparse Multimodal Data. In *Proceedings of the 32nd ACM International Conference on Multimedia (MM '24), October 28-November 1, 2024, Melbourne, VIC, Australia.* ACM, New York, NY, USA, 10 pages. https://doi.org/10.1145/3664647.3680706

## 1 Introduction

Traffic forecasting plays a crucial role in Intelligent Transportation Systems (ITS), which use historical traffic signals from sensors to predict future traffic signals. Traditional statistics-based methods, such as exponential smoothing [19] and autoregressive integrated moving average [37], have some limitations due to their reliance on stationarity-related assumptions and disregard for nonlinear relationships between traffic signals. Recent research based on deep learning captures traffic patterns from both temporal and spatial perspectives, using TCN-based [26, 39], RNN-based [3, 22, 32], and attention-based [11, 12, 16, 17] modules to model the temporal correlation and GCN-based modules [9, 21, 31, 40] to model the spatial correlation between traffic nodes. Currently, traffic forecasting has made substantial advancements and is utilized in several downstream applications, such as traffic management, urban computing, and automated driving [2, 14, 41].

However, various human activities that occur in urban areas have impacts on the city transportation system, leading to abrupt shifts in traffic patterns. Consequently, the traffic prediction model with excellent regular performance is weak in this case as shown in Figure. 1. Events, such as sports championships, exhibits, and concerts, are typical large-scale human activities that have the characteristic of impacting wide areas and attracting large crowds, pose serious challenges to event organizers and traffic management agencies. Inaccurate estimation of the event's influence on the traffic system may result in traffic congestion in the surrounding area. In some extreme cases, it can result in safety concerns such as traffic accidents and crowd surge. Hence, this study specifically addresses the issue of event traffic forecasting, with the goal of

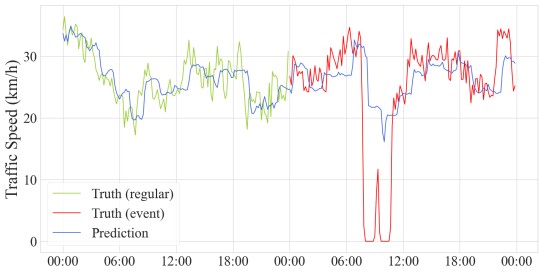

**Figure 1: The impact of events on traffic patterns can lead to distortions in traffic prediction models. GWNet is a powerful model that works well when no events occur (regular), but weakens when events occur.**

modeling the influence of events on traffic patterns and effectively predicting event traffic.

In this paper, we highlight that the primary challenges of event traffic forecasting research are the **diversity** and **sparsity** of events after analyzing the real-world traffic data in Shenzhen City, China. First, there are various kinds of events held in cities, including for-profit, non-profit, official, unofficial, popular, and professional ones. Due to the diversity of events, it is challenging for researchers to construct a complete set of features for uniformly representing the events' characteristics, which makes analyzing the impact of events on traffic patterns difficult. Second, the spatial and temporal distribution of events is sparse. For example, no more than half of the days are allocated for hosting events at the Shenzhen Convention and Exhibition Center. Furthermore, the amount of data is smaller during certain time ranges, such as weekdays or special holidays, bringing a huge challenge for data-driven methods to model spatialtemporal patterns.

In addition to the inherent challenges presented by the data itself, the lack of enough data also greatly limits the progress of this research. Existing public traffic forecasting datasets, such as METR-LA, PEMS-BAY, and PEMS0X, only contain the time-series modal traffic signal data and the graph-style node connection matrix, without any event-related information. Since there is no official website to directly download accurate and comprehensive event information, we can only obtain it from the Internet, where useful data is scattered in every corner. Manually retrieving relevant web pages through search engines to create datasets is time-consuming and not feasible for real-world applications. We have to solve the problem of dataset construction pipeline before developing the event forecasting method.

Therefore, addressing the above issues to effectively leverage multimodal data is essential for improving the performance of event traffic forecasting. To achieve this, we first propose an efficient event description text generation pipeline with large language models (LLMs). We send specially designed prompts and retrieved related web pages to the powerful LLM for generating textual descriptions for events. We use this data generation pipeline to build two multi-modal event traffic prediction dataset at the Shenzhen Convention and Exhibition Center (ShenzhenCEC) and Suzhou International Exp Centre (SuzhouIEC). Then, we design a multimodal (**T**ext and **T**ime-series) event **T**raffic forecasting model, T3, which

uses pre-trained text and traffic encoders to extract embeddings of corresponding modalities for prediction. The text and traffic encoders, which are trained on a large amount of data, have strong generalization capabilities to learn the representation of sparse data without additional training. Learnable projections are used to transform embeddings into hidden features to adjust for the prediction task, followed by two multi-layer perceptrons for outputting the prediction results together from the hidden features.

We summarize key contributions of this paper as follows:

- We first propose the event forecasting problem in a multimodal setting, which uses event description text and historical traffic data to predict future traffic signals.
- We develop an LLM-based event description text generation pipeline, which significantly reduces the cost of obtaining event information from websites. And we build two multimodal event traffic forecasting datasets.
- We propose the multimodal event traffic forecasting model, T3, which uses the generalization capabilities of pre-trained encoders to obtain representations of sparse data.
- We conduct extensive experiments on ShenzhenCEC and SuzhouIEC to gain insight into the effectiveness of the T3. Experimental results show that our proposal is able to consistently and significantly outperform all baselines and reduce the MAE by 4.26% during the event peak period.

## 2 Related Work

### 2.1 Traffic Forecasting

Traffic forecasting is an important type of time series forecasting. Previous studies treated the traffic forecasting problem as a pure time series prediction task and addressed it via traditional statistic-based methods, such as exponential smoothing[19] and autoregressive integrated moving average [37]. These methods rely heavily on stationarity-related assumptions and ignore the nonlinear correlations between traffic signals, which severely limits traffic forecasting's effectiveness. Recently, deep learning-based studies have been proposed to capture the complex spatial-temporal correlations in traffic signals. DCRNN [22] and STGCN [42] were the first to apply deep learning methods to traffic prediction, using GCN to capture spatial correlation and RNN or CNN to capture temporal correlation, and have made significant progress. Subsequently, more and more methods [10, 20, 44] design exquisite spatiotemporal feature extraction modules to achieve better performance. Recently, some works about time-series or traffic fundamental models [15, 29, 47] use time-series data to fine-tune pre-trained LLMs to obtain zero-shot prediction capabilities. However, it is worth noting that these studies have not utilized multi-modal data.

### 2.2 Multimodal Machine Learning

Multimodal machine learning aims to develop models that can process and relate information from multiple modalities, such as image, video, audio, and 3D [4]. Existing multi-modal learning research mainly focuses on vision, audio, and language modal, and a series of research works have emerged, such as SUR-adapter [46], RTQ [36], HAT [5], PromptMTopic [28], and so on [23]. And there are also some papers focus on the 3D point cloud, table, source code, graph, etc [6, 13, 33, 35]. There are many core technical challenges

**Table 1: Frequently used notation.**

| Notation | Description |
|---|---|
| $S$ | Set of traffic nodes |
| $E$ | Set of traffic edges |
| $N$ | Number of traffic nodes |
| $M$ | Number of traffic edges |
| $T_h$ | Length of historical time steps |
| $T_f$ | Length of future time steps |
| $X_i$ | Traffic signal of the $i$-th time step |
| $\mathcal{X}$ | Traffic signal of the $T_h$ most recent past time steps |
| $\mathcal{G}$ | Traffic network $\mathcal{G} = (V, E)$ |
| $\mathcal{T}$ | Daily event description text |
| $\mathcal{Y}$ | Traffic signal of the $T_f$ nearest future time steps |
| $D^{\text{text}}$ | The output dimension of text encoder |
| $D^{\text{traffic}}$ | The output dimension of traffic encoder |
| $D$ | The hidden dimension of the T3 model |

**Table 2: Data statistics.**

| City | Node | Edge | Interval | Time Steps | Event Day |
|---|---|---|---|---|---|
| Shenzhen | 742 | 1277 | 10 min | 18,000 | 44 |
| SuZhou | 8 | 16 | 10 min | 52,416 | 30 |

surrounding multimodal machine learning, and we focus on multimodal representation and fusion in this paper. Representation involves learning how to represent and summarize multimodal data using complementarity and redundancy of multiple modalities. Pre-trained encoders [7, 18, 38] are commonly used to learn representations of inputs from relevant modalities. Fusion involves joining information from two or more modalities to perform a prediction. There are many model-agnostic approaches and model-based approaches [1]. With the development of large language models, multimodal large models are also attracting more and more attention, such as BLIP-2 [43], LLaVA [25], and GPT4V.

## 3 Preliminaries

In this section, we define the notions of traffic sensor, network, signal, and the event traffic forecasting problem. Table 1 shows the frequently used notation.

**Traffic Sensor**. A traffic sensor is a sensor deployed in a traffic system that records traffic states such as the flow or speed of passing vehicles.

**Traffic Network**. A traffic network is defined as a directed or undirected graph with the formula $\mathcal{G} = (S, E)$, where $S$ is the set of $|S| = N$ traffic nodes and $E$ is the set of $|E| = M$ edges.

**Traffic Signal**. The traffic signal $X_t \in \mathbb{R}^N$ denotes the observation of all sensors in the traffic network $\mathcal{G}$ at time step $t$. In this paper, the traffic signals generally means the traffic speed.

**Event Description Text**. Event description text $\mathcal{T}$ is a text that describes the main contents of the events in a specific time period.

**Event Traffic Forecasting**. Given historical traffic signal $\mathcal{X} = [X_{t-T_h}, X_{t-T_h+1}, \cdots, X_{t-1}] \in \mathbb{R}^{T_h \times N}$ from the past $T_h$ time steps, traffic network $\mathcal{G}$ and event description text $\mathcal{T}$, the event traffic forecasting can be formed as:

$$\hat{\mathcal{Y}} = \mathcal{F}(\mathcal{X}, \mathcal{G}, \mathcal{T}),$$

where $\mathcal{Y} \in \mathbb{R}^{T_f \times N}$ are the nearest future $T_f$ time steps of traffic signal, and $\mathcal{F}$ is the forecasting model. $\mathcal{T}$ can be $\varnothing$ when there is no event occurring, and this situation is equivalent to the classical traffic forecasting problem.

## 4 Dataset

In this section, we will introduce in detail the construction process and analysis conclusions of the multi-modal event traffic forecasting datasets, using ShenzhenCEC as an example.

### 4.1 Event Description Text Generation Pipeline

Urban areas often hold a diverse range of large-scale events, including exhibitions, concerts, sports matches, charity galas, etc. These events attract large numbers of people and have a significant impact on the surrounding traffic pattern, making it different from the normal state. We try to collect comprehensive information about these events to enhance the performance of event traffic prediction. However, there is currently no official and unified way to obtain event information, and comprehensive event details are scattered in every corner of the Internet. We find that some urban venues' official websites include historical event schedule information. Nevertheless, this tabular data is limited to the event's name and period, lacking comprehensive details such as the event's introduction, theme, anticipated viewership, and so on. Although researchers can use search engines (like Google) to retrieve relevant data from massive web pages and manually construct a toy dataset, this data collection approach is time-consuming and labor-intensive, making it challenging to apply in real-world scenarios.

Hence, we develop an event description text generation pipeline based on the advanced large language model as depicted in Figure. 2. There are four steps to generate a text for the event. First, event scheduling data is downloaded from the official website of venues (such as the homepage of Shenzhen Convention and Exhibition Center [1]), which only includes the name and time of historical events. Then, we create a prompt that adequately describes the requirements of our goal, where the prompt should contain the name and time of the event to alleviate hallucinations of LLM's. Next, we crawl event-related webpages from the Internet (or you can use similar tools provided by commercial services directly) as the raw material for LLMs to generate text. Finally, we send both the prompt and web data to a LLM to generate the event description text. This retrieval augmented generation (RAG) pipeline can automatically retrieve event-related information from the Internet and intelligently extract text summaries. Compared with the manual pipeline, our proposed solution is more convenient, efficient, and low-cost, which can be extended to real-life scenarios.

It is crucial to acknowledge that the data generated by the aforementioned pipeline may be at **data leakage** risk for the forecasting task, meaning that information about upcoming incidents or emergencies may be included in the event description text. This potential risk can be resolved by limiting the publication time of crawled web pages, making the LLM only obtain web data published before

---

[1] https://www.ShenzhenCEC.com/

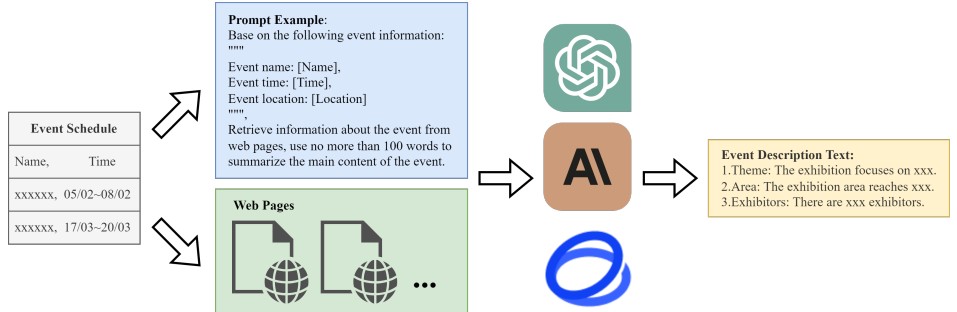

**Figure 2: The event description text generation pipeline. We gather event scheduling data from official websites, create a prompt for LLMs to generate event description text, crawl event-related webpages, and send both data and prompts to the LLM for generating the event description.**

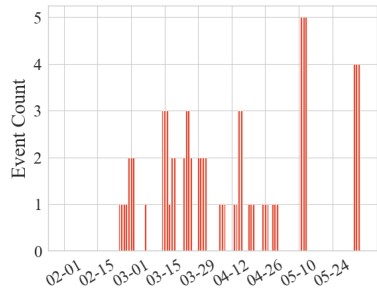

**Figure 3: Distribution of events in time.**

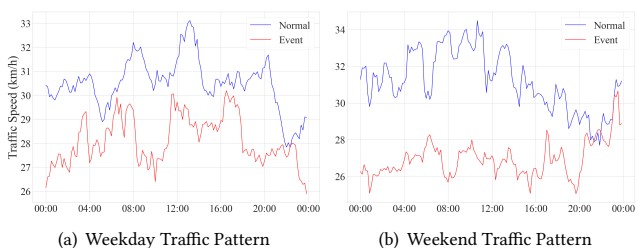

(a) Weekday Traffic Pattern    (b) Weekend Traffic Pattern

**Figure 4: The impact of the event on weekday or weekend traffic patterns. The blue line depicts the average traffic speed during days without any events, whereas the red line illustrates the average traffic speed during days with events.**

the event starts. Another issue is **the hallucination of LLM**. Although several prompt engineering tricks, such as system prompts and few-shot prompts, can significantly reduce the occurrence of hallucinations, it is nearly impossible to entirely eliminate this problem. In order to ensure the quality of the data, we manually check all event description texts to ensure that there is no information leakage or obvious hallucinatory context. Following the above pipeline, we generate description texts for all events held at the Shenzhen Convention and Exhibition Center between the dates of February 1st, 2023, and June 5th, 2023.

## 4.2 Data Statistics

We select the Shenzhen Convention and Exhibition Center (ShenzhenCEC) to build the dataset because of its high frequency of large-scale events. The statistical information is shown in Table. 2. **The traffic speed data** is collected at regular intervals of 10 minutes within a radius of 1000 meters surrounding the ShenzhenCEC. The time range corresponds to the duration of the event data, spanning a total of 125 days from February 1st, 2023, to June 5th, 2023. **The traffic graph** is constructed based on the node distances. We calculate the distance between nodes and only connect node pairs that are within a distance of less than 10 meters. The traffic data consists of a total of 18,000 time steps, 742 traffic nodes (road links), and 1277 edges. The Figure. 3 illustrates the **temporal distribution of events** at the ShenzhenCEC over a span of 125 days. This location hosted a total of 34 events, which were distributed across 44 days, accounting for 35.2% of the total days. It is important to note that events can last over multiple days, and it is possible for multiple events to occur within a single day. A single day can have a maximum of 5 events, and a single event can last up to 4 days in our dataset. The occurrence of events is observed on 28 weekdays and 16 weekends.

## 4.3 Impact of Event on Traffic Pattern

Figure. 4 (a) and (b) illustrate the average traffic speed on weekdays and weekends respectively. The blue line indicates the average traffic speed during days without any events, while the red line represents the average traffic speed during days with events. We can conclude that the events can consistently decrease traffic speed throughout the entire nearby area. On weekdays, there are significant variations in traffic patterns at four time periods: nighttime, around 10 a.m., around 2 p.m., and the evening rush hours. The traffic speed reduction at night primarily stems from the organizers' requirement to prearrange the booth, resulting in a substantial influx and outflow of vehicles. The time periods around 10 a.m. and 2 p.m. are generally the start times of exhibition-type events, where the concentration of exhibitors and visitors leads to traffic congestion. Finally, the commuting traffic flow during the evening peak period merges with the traffic flow after the exhibition, causing the evening peak hour to arrive earlier and last longer. During the weekends, the absence of commuter traffic results in a relatively

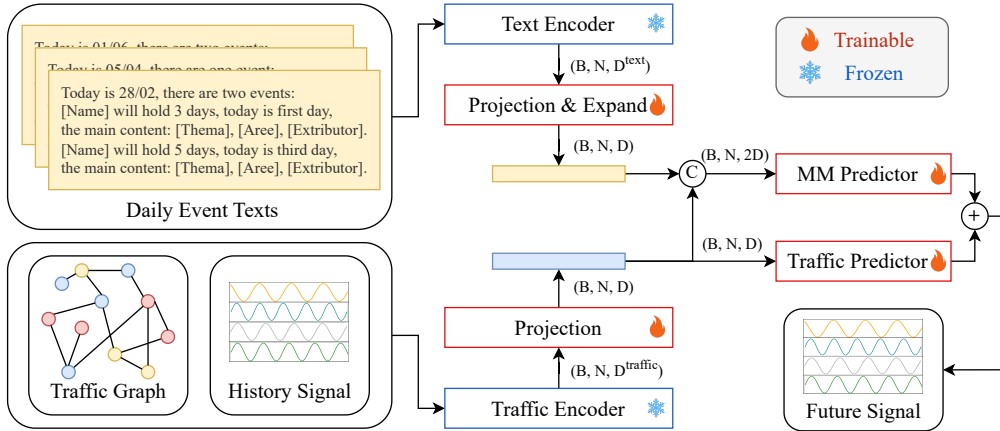

**Figure 5: The framework of our proposed T3. Modules in the blue box with *snowflake* mean frozen in the training stage, while in the red box with *flame* mean trainable. $C$ is the concat operation, and + is the sum operation.**

stable average traffic speed. Events cause a general reduction in traffic speed, while the speed remains mostly unchanged after 20 o'clock, regardless of whether an event takes place.

## 5 T3: Multimodal Forecasting Model

### 5.1 Overview

As shown in Figure. 5, we propose a multimodal (**T**ext and **T**ime-series) **T**raffic forecasting model, T3. This model uses a pre-trained text encoder and traffic encoder to extract the embedding of the corresponding modal data. The generalization ability of pre-trained modules can deal with the challenges of sparse event data. Then, the T3 projects and fuses the two embeddings to obtain a multimodal feature. Finally, two independent predictors output intermediate prediction results from the multi-modal feature and traffic feature, and their outputs are added as the final prediction result. Text and traffic encoders are frozen in the training stage (in the blue box with *snowflake*), and other modules are trainable (in the red box with *flame*). Instead of designing a sophisticated model, we propose a straightforward way to verify whether multi-modal data can improve the performance of the event traffic forecasting.

### 5.2 Model Structure

*Frozen Text Encoder.* We use the pre-trained text embedding model as the text encoder to obtain the text representations for event traffic forecasting. Text embedding [34] is a commonly used NLP technique that converts text data into fixed-dimensional vectors that can be processed by machine learning/deep learning algorithms. These vector representations are designed to capture the semantics and context of the words they represent. We combine the texts of multiple events that occurred in a day as the daily event description text $\mathcal{T}$ and send it into the text encoder to get text embedding $\text{emb}^{\text{text}} \in \mathbb{R}^{B \times D^{\text{text}}}$:

$$\text{emb}^{\text{text}} = \text{TextEncoder}(\mathcal{T}). \qquad (1)$$

We freeze the text encoder parameters in the training stage to keep the powerful generalization ability learned from massive text training data for sparse event text embedding.

*Frozen Traffic Encoder.* We use the encoder layers of the traffic forecasting model, which is pre-trained on traffic data from the traffic dataset, as the traffic encoder. Common traffic forecasting models use time-series data (history traffic signals) and graph-structure data (traffic network) as inputs. These methods often employ space operations, such as graph convolutional networks (GCN) and spatial attention, to capture the spatial correlation among traffic nodes, and sequence operations, such as TCN, attention, and recurrent neural networks (RNN), to capture the temporal correlation. Our proposed T3 uses the spatiotemporal modules as the traffic encoder to extract traffic embedding $\text{emb}^{\text{traffic}} \in \mathbb{R}^{B \times N \times D^{\text{traffic}}}$:

$$\text{emb}^{\text{traffic}} = \text{TrafficEncoder}(\mathcal{X}, \mathcal{G}). \qquad (2)$$

*Project, Expand, and Concat Embeddings.* Although the pre-trained frozen encoders have strong generalization capabilities to extract sparse text and traffic embeddings, it is necessary for our model to maximize the utilization of the training data to adjust the embeddings to fit the event traffic forecasting task. First, we use two layers MLP to project the embeddings into the hidden space independently:

$$\mathbf{H}^{\text{text}} = \text{MLP}_{\text{text}}(\text{emb}^{\text{text}}), \mathbf{H}^{\text{traffic}} = \text{MLP}_{\text{traffic}}(\text{emb}^{\text{traffic}}), \qquad (3)$$

where the shape of $\mathbf{H}^{\text{text}}$ is $(B, D)$ and the shape of $\mathbf{H}^{\text{traffic}}$ is $(B, N, D)$. Then, we extend the size of $\mathbf{H}^{\text{text}}$ to $(B, N, D)$ and concatenate the two hidden features to get multi-modal features $\mathbf{H}^{\text{mm}} \in \mathbb{R}^{B \times N \times 2D}$:

$$\mathbf{H}^{\text{mm}} = \mathbf{H}^{\text{text}} \parallel \mathbf{H}^{\text{traffic}}. \qquad (4)$$

*Predictor and Loss.* We apply regression layers on $\mathbf{H}^{\text{mm}}$ and $\mathbf{H}^{\text{traffic}}$ respectively, and add their output results as the final prediction result:

$$\hat{\mathcal{Y}} = \text{FC}_2^{\text{mm}}(\theta(\text{FC}_1^{\text{mm}}(\mathbf{H}^{\text{mm}}))) + \text{FC}_2^{\text{traffic}}(\theta(\text{FC}_1^{\text{traffic}}(\mathbf{H}^{\text{traffic}}))), \quad (5)$$

where $\text{FC}(\cdot)$ is fully connection layer and $\theta(\cdot)$ is activation function. In the training stage, we use Mean Absolute Error (MAE) as loss function:

$$\mathcal{L} = \text{MAE}(\mathcal{Y}, \hat{\mathcal{Y}}). \qquad (6)$$

**Table 3: Traffic forecasting results on the ShenzhenCEC and SuzhouIEC dataset during the overall period and event peak hours. Black bold indicates the best result, and underlining indicates the second-best result.**

| | Model | Overall | | | | Event Peak Hours | | | |
|---|---|---|---|---|---|---|---|---|---|
| | | MAE | RMSE | MAPE (%) | WAPE (%) | MAE | RMSE | MAPE (%) | WAPE (%) |
| ShenzhenCEC | HI | 8.09±0.000 | 10.33±0.000 | 32.07±0.000 | 58.55±0.000 | 8.24±0.000 | 11.32±0.000 | 36.05±0.000 | 57.88±0.000 |
| | NLinear | 4.42±0.006 | 6.80±0.004 | 16.40±0.023 | 16.00±0.023 | 5.85±0.004 | 8.76±0.021 | 23.91±0.014 | 20.57±0.013 |
| | PatchTST | 4.31±0.006 | 6.70±0.016 | 16.04±0.019 | 15.60±0.022 | 5.65±0.017 | 8.67±0.083 | 23.16±0.102 | 19.84±0.061 |
| | DCRNN | 3.84±0.055 | 5.87±0.120 | 14.04±0.140 | 13.87±0.198 | 4.80±0.149 | 6.70±0.250 | 20.80±0.493 | 16.87±0.524 |
| | GWNet | _3.69_±0.019 | _5.68_±0.064 | _13.68_±0.272 | _13.35_±0.067 | _4.46_±0.169 | 6.65±0.150 | _19.98_±0.108 | _15.66_±0.595 |
| | D2STGNN | 3.89±0.121 | 6.13±0.317 | 14.32±0.402 | 14.08±0.437 | 4.82±0.227 | 7.30±0.489 | 21.20±0.876 | 16.94±0.799 |
| | STID | 4.02±0.066 | 5.99±0.112 | 14.95±0.216 | 14.54±0.238 | 4.84±0.148 | 7.01±0.139 | 20.98±0.415 | 16.99±0.520 |
| | STAEformer | 3.80±0.109 | 5.76±0.153 | 14.03±0.436 | 13.73±0.393 | 4.50±0.216 | _6.57_±0.457 | 20.22±0.939 | 15.81±0.758 |
| | T3 (D=128) | 3.73±0.017 | 5.74±0.096 | 13.85±0.170 | 13.49±0.060 | 4.46±0.122 | 6.72±0.132 | 20.06±0.338 | 15.68±0.430 |
| | T3 (D=256) | **3.66**±0.033 | **5.51**±0.116 | **13.62**±0.165 | **13.24**±0.121 | **4.27**±0.154 | **6.31**±0.137 | **19.23**±0.356 | **15.00**±0.542 |
| | T3 (D=512) | 3.71±0.133 | 5.70±0.349 | 13.86±0.423 | 13.43±0.480 | 4.42±0.458 | 6.65±0.608 | 19.94±1.462 | 15.52±1.610 |
| SuzhouIEC | HI | 5.31±0.000 | 8.32±0.000 | 19.43±0.000 | 18.88±0.000 | 6.59±0.000 | 9.95±0.000 | 26.87±0.000 | 23.15±0.000 |
| | NLinear | 4.01±0.002 | 6.14±0.011 | 14.86±0.010 | 14.51±0.005 | 4.48±0.013 | 6.47±0.020 | 19.13±0.051 | 15.77±0.045 |
| | PatchTST | 3.52±0.010 | 5.18±0.024 | 13.55±0.018 | 13.28±0.022 | 3.90±0.006 | 5.53±0.025 | 17.53±0.021 | 13.75±0.022 |
| | DCRNN | 3.25±0.056 | 4.86±0.158 | 11.90±0.122 | 12.02±0.116 | 3.00±0.052 | 4.27±0.061 | 13.38±0.230 | 10.57±0.181 |
| | GWNet | 3.34±0.084 | 5.13±0.182 | 11.94±0.187 | 12.06±0.158 | 2.94±0.028 | 4.20±0.038 | 12.83±0.140 | 10.38±0.097 |
| | D2STGNN | 3.24±0.060 | 4.93±0.166 | 11.40±0.162 | 11.74±0.134 | 2.67±0.040 | 3.76±0.044 | **11.34**±0.074 | 9.41±0.141 |
| | STID | 3.10±0.029 | 4.56±0.060 | 11.08±0.060 | 11.47±0.064 | **2.66**±0.018 | 3.76±0.032 | 11.66±0.042 | 9.42±0.064 |
| | STAEformer | 3.12±0.047 | 4.67±0.058 | 11.18±0.146 | 11.37±0.164 | 2.78±0.026 | 3.91±0.027 | 12.08±0.217 | 9.41±0.091 |
| | T3 (D=128) | 3.04±0.011 | 4.49±0.011 | 10.88±0.021 | 11.26±0.033 | **2.66**±0.007 | **3.74**±0.004 | 11.44±0.010 | _9.39_±0.023 |
| | T3 (D=256) | _3.03_+0.014 | _4.47_±0.015 | _10.85_±0.021 | _11.21_±0.035 | **2.66**±0.011 | **3.74**±0.009 | _11.45_±0.019 | **9.38**±0.040 |
| | T3 (D=512) | **3.02**±0.012 | **4.46**±0.011 | **10.84**±0.018 | **11.19**±0.031 | **2.66**±0.013 | _3.75_±0.013 | 11.47±0.014 | _9.39_±0.044 |

## 6 Experiment

### 6.1 Experimental Setting

*6.1.1 Dataset.* All experiments are performed on the proposed two multimodal event traffic forecasting datasets described in Section. 4. The datasets are chronologically divided into training, validation, and testing as 6:2:2. Following the common setting in previous traffic forecasting works [22], which uses the traffic signals of the history $T_f$ = 12 time steps to predict the next $T_h$ = 12 steps. Event peak hours are 7 a.m. to 10 a.m. and 5 p.m. to 8 p.m. during days with events.

*6.1.2 Metrics.* We evaluate the performances of all baselines by four commonly used metrics in traffic forecasting, including mean absolute error (MAE), root mean square error (RMSE), mean absolute percentage error (MAPE), and weight mean absolute percentage error (WAPE). MAE reflects prediction accuracy; RMSE is more sensitive to abnormal values; MAPE can eliminate the influence of data units to some extent; and WAPE is more robust to outliers compared with MAPE.

*6.1.3 Implementation Details.* All traffic forecasting models are implemented using pytorch 1.10 and cudn 11.3. Models are trained and evaluated on 12th Gen Intel(R) Core(TM) i9-12900K, 128 GB RAM computing server equipped with two RTX 4090 GPUs. We employ Adam with a learning rate of 0.002 as our optimizer. We use the multistep learning rate scheduling strategy and set the decay ratio to 0.5. All models are repeated three times using fixed seed 0, 1, and 2. We use voyage-2 to get text embeddings with $D^{\text{text}}$ = 1024

and GWNet to get traffic embeddings with $D^{\text{traffic}}$ = 256. Other implementation details can be seen in the code.

*6.1.4 Baselines.* We compare our proposed T3 model with the eight baseline models:

- **HI** [8]: Historical Inertia simply uses the most recent data points in the input time series as predictions.
- **NLinear** [45]: NLinear is a MLP-based time series forecasting model that decomposes the time series into a trend and a remainder series and employs two parallel linear layers to predict these two series.
- **PatchTST** [27]: PatchTST is a transformer-based time-series forecasting model, which divides the input time series data into subseries-level patches which are served as input tokens to Transformer.
- **DCRNN** [22]: Diffusion convolution recurrent neural network is an encoder-decoder structure network that combines graph convolution networks with RNN.
- **GWNet** [39]: Graph WaveNet jointly captures spatial and temporal dependencies through the sequential integration of temporal convolution layers and graph convolutional layers.
- **D2STGNN** [32]: This method integrates deep learning technology and signal processing theory to enhance the accuracy and efficiency of predictions.
- **STID** [30]: STID uses learnable temporal embeddings and node embeddings to model spatialtemporal correlation. This simple yet efficient component allows it to achieve performance comparable to SOTA through multiple MLP layers.

**Table 4: Traffic forecasting results on the ShenzhenCEC dataset during the overall period and event peak hours at the** 3**-rd,** 6**-th and** 12**-th time steps.** *Step* **means time step.**

| Model | Step | Overall | | | | Event Peak Hours | | | |
|---|---|---|---|---|---|---|---|---|---|
| | | MAE | RMSE | MAPE (%) | WAPE (%) | MAE | RMSE | MAPE (%) | WAPE (%) |
| HI | 3 | 8.10±0.000 | 10.34±0.000 | 32.10±0.000 | 58.60±0.000 | 8.24±0.00 | 11.32±0.000 | 36.05±0.000 | 57.88±0.000 |
| | 6 | 8.09±0.000 | 10.33±0.000 | 32.08±0.000 | 58.56±0.000 | 8.24±0.000 | 11.32±0.000 | 36.05±0.000 | 57.88±0.000 |
| | 12 | 8.09±0.000 | 10.32±0.000 | 32.03±0.000 | 58.46±0.000 | 8.24±0.000 | 11.32±0.000 | 36.05±0.000 | 57.88±0.000 |
| NLinear | 3 | 3.42±0.005 | 5.33±0.003 | 12.83±0.013 | 12.37±0.018 | 4.42±0.014 | 7.11±0.019 | 18.55±0.021 | 15.54±0.047 |
| | 6 | 4.39±0.002 | 6.77±0.004 | 16.26±0.011 | 15.88±0.009 | 5.91±0.009 | 8.95±0.008 | 23.93±0.004 | 20.75±0.030 |
| | 12 | 5.70±0.017 | 8.31±0.008 | 20.98±0.063 | 20.60±0.061 | 7.32±0.018 | 10.19±0.063 | 29.62±0.101 | 25.73±0.063 |
| PatchTST | 3 | 3.31±0.012 | 5.18±0.015 | 12.50±0.034 | 11.99±0.044 | 4.22±0.045 | 6.83±0.022 | 17.82±0.222 | 14.82±0.158 |
| | 6 | 4.27±0.014 | 6.65±0.035 | 15.86±0.044 | 15.45±0.051 | 5.65±0.032 | 8.81±0.164 | 23.08±0.063 | 19.85±0.112 |
| | 12 | 5.59±0.017 | 8.22±0.008 | 20.62±0.039 | 20.21±0.063 | 7.16±0.040 | 10.15±0.030 | 29.12±0.096 | 25.14±0.142 |
| DCRNN | 3 | 3.03±0.038 | 4.63±0.056 | 11.39±0.073 | 10.98±0.139 | 3.75±0.059 | 5.62±0.155 | 16.99±0.193 | 13.19±0.208 |
| | 6 | 3.83±0.054 | 5.85±0.090 | 14.11±0.072 | 13.86±0.195 | 4.81±0.138 | 6.79±0.224 | 21.14±0.405 | 16.89±0.484 |
| | 12 | 4.85±0.128 | 7.16±0.230 | 17.29±0.448 | 17.53±0.462 | 6.00±0.313 | 7.68±0.417 | 24.58±1.062 | 21.08±1.099 |
| GWNet | 3 | 2.86±0.032 | 4.47±0.050 | 10.81±0.067 | 10.36±0.114 | 3.45±0.134 | 5.49±0.036 | 16.09±0.120 | 12.12±0.470 |
| | 6 | 3.66±0.037 | 5.70±0.126 | 13.63±0.220 | 13.25±0.134 | 4.47±0.087 | 6.82±0.215 | 20.45±0.152 | 15.72±0.306 |
| | 12 | 4.76±0.165 | 6.92±0.269 | 17.39±1.002 | 17.20±0.597 | 5.57±0.120 | 7.68±0.422 | 23.92±0.500 | 19.58±0.423 |
| D2STGNN | 3 | 2.89±0.059 | 4.48±0.085 | 10.90±0.241 | 10.45±0.214 | 3.49±0.110 | 5.49±0.125 | 15.90±0.420 | 12.25±0.386 |
| | 6 | 3.79±0.103 | 5.89±0.119 | 14.04±0.500 | 13.70±0.371 | 4.75±0.214 | 7.25±0.310 | 21.48±0.797 | 16.67±0.750 |
| | 12 | 5.31±0.486 | 7.92±0.905 | 19.18±1.408 | 19.20±1.756 | 6.56±0.886 | 9.09±1.236 | 27.37±3.040 | 23.06±3.113 |
| STID | 3 | 3.23±0.083 | 4.87±0.160 | 12.14±0.207 | 11.68±0.301 | 3.93±0.185 | 6.00±0.330 | 17.13±0.339 | 13.81±0.651 |
| | 6 | 4.00±0.075 | 5.99±0.125 | 14.93±0.238 | 14.48±0.270 | 4.91±0.201 | 7.17±0.213 | 21.53±0.685 | 17.24±0.705 |
| | 12 | 5.02±0.113 | 7.14±0.150 | 18.44±0.532 | 18.13±0.409 | 5.76±0.222 | 7.82±0.282 | 24.66±0.713 | 20.22±0.778 |
| STAEformer | 3 | 3.00±0.042 | 4.55±0.033 | 11.30±0.341 | 10.86±0.152 | 3.59±0.190 | 5.46±0.115 | 16.80±0.496 | 12.61±0.667 |
| | 6 | 3.79±0.126 | 5.76±0.133 | 14.14±0.683 | 13.72±0.456 | 4.47±0.156 | 6.69±0.428 | 20.54±0.729 | 15.70±0.549 |
| | 12 | 4.77±0.271 | 6.94±0.348 | 17.31±0.790 | 17.25±0.978 | 5.57±0.468 | 7.47±0.651 | 23.74±1.859 | 19.58±1.643 |
| T3 (Ours) | 3 | 2.85±0.013 | 4.37±0.052 | 10.78±0.037 | 10.31±0.045 | 3.41±0.075 | 5.35±0.059 | 16.05±0.183 | 11.98±0.262 |
| | 6 | 3.64±0.019 | 5.50±0.131 | 13.61±0.162 | 13.17±0.067 | 4.28±0.086 | 6.48±0.213 | 19.73±0.258 | 15.05±0.303 |
| | 12 | 4.62±0.122 | 6.58±0.166 | 16.96±0.579 | 16.71±0.441 | 5.09±0.301 | 6.82±0.348 | 21.91±1.074 | 17.90±1.058 |

- **STAEformer** [24]: Spatiotemporal adaptive embedding transformer proposes a novel component called spatiotemporal adaptive embedding that can yield outstanding results with vanilla transformers.

Among these methods, HI, NLinear, and PatchTST are time series prediction models, and the others are traffic prediction models.

## 6.2 Performance Comparison

To verify the generality and performance of our proposed T3 model, we compare it with eight baselines. Table. 3 shows the interval estimate results of average forecasting performance. The HI algorithm has the worst performance because it cannot capture complex traffic patterns in both time and space dimensions. Although NLinear and PatchTST are widely used in the time series forecasting task, they are significantly inferior to traffic forecasting methods due to their lack of particular modules for learning spatial correlation inside the traffic network. Among the five traffic forecasting baselines, GWNet demonstrates almost the best performance, which may be attributed to its gated structure that can learn time-varying changes in traffic speed, while STAEformer achieves the second highest ranking results, possibly contributed by its transformer

structure that can learn complex traffic patterns. Compared with the above eight baseline methods, our proposed T3 model gets state-of-the-art performance across all metrics, especially during peak hours (7 a.m. to 10 a.m. and 5 p.m. to 8 p.m.) on event days, achieving a reduction in MAE of 4.26%. Furthermore, Table. 4 shows in detail the prediction performance of all methods at the third (half an hour), sixth (an hour), and twelfth (two hours) time steps. Our method demonstrates strong performance across all three time steps. In summary, our method can consistently and significantly outperform eight strong baselines and reduce the MAE by 4.26% during the event peak period.

## 6.3 The Efficacy of Text Embedding

We conduct two groups of experiments and text embedding visualization to verify the effectiveness of text embedding for the event traffic forecasting task. First, we discuss whether multimodal data is necessary for the event traffic forecasting problem. We conduct a comparison between the GWNet employing handcrafted event features and our proposed approach utilizing text embeddings. The original inputs of GWNet have three kinds of features: traffic speed, time of day, and day of week. We design two extra event features:

**Table 5: Comparison learning event embeddings with hand-craft event features. *P* is period, where *All* represents overall period and *EP* represents event peak hours. *EO* is the event occurring feature, while *EN* is the event number feature.**

| P | Type | MAE | RMSE | MAPE (%) | WAPE (%) |
|---|------|-----|------|----------|----------|
| All | Naive | $3.69_{\pm0.019}$ | $5.68_{\pm0.064}$ | $13.68_{\pm0.272}$ | $13.35_{\pm0.067}$ |
| | +EO | $3.73_{\pm0.015}$ | $5.73_{\pm0.029}$ | $13.90_{\pm0.109}$ | $13.50_{\pm0.056}$ |
| | +EN | $3.76_{\pm0.052}$ | $5.79_{\pm0.184}$ | $13.96_{\pm0.211}$ | $13.60_{\pm0.189}$ |
| | Ours | $3.66_{\pm0.033}$ | $5.51_{\pm0.116}$ | $13.62_{\pm0.165}$ | $13.24_{\pm0.121}$ |
| EP | Naive | $4.46_{\pm0.169}$ | $6.65_{\pm0.150}$ | $19.98_{\pm0.108}$ | $15.66_{\pm0.595}$ |
| | +EO | $4.37_{\pm0.071}$ | $6.57_{\pm0.045}$ | $19.71_{\pm0.047}$ | $15.36_{\pm0.249}$ |
| | +EN | $4.66_{\pm0.155}$ | $6.76_{\pm0.346}$ | $20.52_{\pm0.612}$ | $16.38_{\pm0.543}$ |
| | Ours | $4.27_{\pm0.154}$ | $6.31_{\pm0.137}$ | $19.23_{\pm0.356}$ | $15.00_{\pm0.542}$ |

**Table 6: Predictor Ablation study during event peak hours.**

| | MAE | RMSE | MAPE (%) | WAPE (%) |
|---|-----|------|----------|----------|
| T3 | $4.27_{\pm0.154}$ | $6.31_{\pm0.137}$ | $19.23_{\pm0.356}$ | $15.00_{\pm0.542}$ |
| w/o MMP | $4.35_{\pm0.096}$ | $6.51_{\pm0.136}$ | $19.58_{\pm0.304}$ | $15.28_{\pm0.339}$ |
| w/o TP | $4.39_{\pm0.098}$ | $6.49_{\pm0.196}$ | $19.66_{\pm0.386}$ | $15.43_{\pm0.346}$ |

**Table 7: Comparison of different types of event embeddings.**

| P | Type | MAE | RMSE | MAPE (%) | WAPE (%) |
|---|------|-----|------|----------|----------|
| All | RE | $3.79_{\pm0.092}$ | $5.86_{\pm0.308}$ | $13.95_{\pm0.479}$ | $13.71_{\pm0.332}$ |
| | GE | $3.78_{\pm0.092}$ | $5.85_{\pm0.327}$ | $13.95_{\pm0.449}$ | $13.67_{\pm0.334}$ |
| | TE | $3.66_{\pm0.033}$ | $5.51_{\pm0.116}$ | $13.62_{\pm0.165}$ | $13.24_{\pm0.121}$ |
| EP | RE | $4.59_{\pm0.360}$ | $6.84_{\pm0.707}$ | $20.40_{\pm1.418}$ | $16.12_{\pm1.265}$ |
| | GE | $4.55_{\pm0.351}$ | $6.83_{\pm0.692}$ | $20.40_{\pm1.418}$ | $15.99_{\pm1.234}$ |
| | TE | $4.27_{\pm0.154}$ | $6.31_{\pm0.137}$ | $19.23_{\pm0.356}$ | $15.00_{\pm0.542}$ |

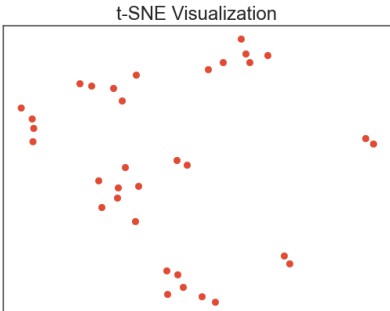

**Figure 6: t-SNE visualization of event embeddings.**

one is whether an event occurs (EO), and the other is the daily events number (EN). Experimental results in Table. 5 show that the EO feature leads to a slight decrease in prediction error during the event peak hours, while the EN feature results in an increase in prediction error throughout both periods. We hypothesize that the EN feature harms model performance because some unique values are absent in the training set but present in the test set. For example, the extreme case of five events in a single day only appears in the test set. Therefore, we can conclude that hand-crafted features struggle to represent the rich context of sparse events, and textual data is necessary for modeling the impact of events on traffic speeds.

Then, we discuss whether the performance improvement comes from the semantic information of text embeddings rather than network structure. We replace the text embeddings (TE) that are output by the text encoder with two kinds of initialized embeddings: one is a random initial embedding (RE), and the other is a Gaussian initial embedding that has the same distribution as TE, namely GE. According to experimental results in Table. 7, TE presents superior performance compared to RE and GE. We can infer that the pre-trained language model can extract information from event text to improve the performance of forecasting. Finally, we use t-SNE technology to reduce high-dimensional text embeddings to two-dimensional formats for visual analysis. As shown in Figure. 6, there are some clusters between text embeddings, which demonstrates that the similarity between events is captured by text embeddings. Similar events will have similar impacts on traffic patterns, and our method obtains the ability to predict event traffic by learning these correlations from historical data.

## 6.4 Ablation and Hyperparameter Study

We conduct the ablation study and hyperparameter study about T3. In Table. 6, *w/o MMP* refers to T3 without the multimodal predictor, and *wo TP* refers to T3 without the traffic predictor. The experimental results indicate that combining two prediction heads yields the best results. We speculate that two predictors can be more effective at learning hidden patterns between events and traffic from sparse data. In addition, Table. 3 presents the impact of different hidden dimensions $D$ on forecasting performance, and the model achieves the best performance when $D$ is set to 256.

## 7 Conclusion

In this paper, we propose the event traffic forecasting problem, which focuses on modeling the impact of events on traffic patterns with multimodal inputs. First, we provide an effective pipeline for generating event text using a large language model and build a multimodal event traffic dataset to describe the diversity of events. Then, to tackle the sparsity of collected data, we propose the multimodal event traffic prediction model, which uses pre-trained text and traffic embedding models to extract embeddings of corresponding modalities for prediction. Finally, we conduct experiments on the real-world dataset, and results show that our proposed method achieves state-of-the-art performance against all baselines.

## Acknowledgments

This work was supported by the National Natural Science Foundation under Grant 62071449 and U20A20184, the Fundamental Research Funds for the Central Universities (E2ET1104), and the Major Key Project of Pengcheng Laboratory, PCL2023A08.

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
