# OpenReview forum: "Event Traffic Forecasting with Sparse Multimodal Data"
_acmmm.org/ACMMM/2024/Conference — MM2024 Poster_

### Official Review · Reviewer_WQWg · 2024-05-24

**Rating:** 2
**Confidence:** 4

**Summary:**

The paper focuses on the problem of event traffic forecasting, which aims to model the impact of various events on traffic patterns in urban areas. The authors highlight the key challenges in this problem, including the diversity and sparsity of events, as well as the lack of sufficient data. To address these challenges, the paper proposes a multi-modal event traffic forecasting model that uses pre-trained text and traffic encoders to extract embeddings from event descriptions and traffic data, and then fuses them for prediction. Additionally, the authors design an efficient LLM-based event description text generation pipeline to build a multi-modal event traffic forecasting dataset called SZCEC. Experimental results on the SZCEC dataset show the proposed method achieves state-of-the-art performance compared to baselines, reducing mean absolute error during event peak periods by 4.26%.

**Strengths:**

1) The paper tackles the novel problem of event traffic forecasting, which is an important but understudied problem in the field of traffic forecasting. The authors identify the unique challenges in this problem and propose a multi-modal approach to address them.
2) The paper Designs an efficient LLM-based event description text generation pipeline to build a new multi-modal event traffic forecasting dataset（SZCEC）, which is valuable for the research community. The use of an LLM-based text generation pipeline to augment the dataset is an innovative approach.
3) The paper provides a thorough evaluation of the proposed method on the SZCEC dataset, comparing it with several baselines. The results demonstrate the effectiveness of the proposed approach, especially during the event peak periods.

**Limitations:**

1) The evaluation is limited to a single dataset, and more comprehensive testing on other real-world event traffic datasets would be needed to better assess the generalizability of the approach.
2) The paper does not provide ablation studies to understand the individual contributions of the different components (e.g., text encoder, traffic encoder) to the overall performance. Such an analysis would help to better understand the strengths and weaknesses of the proposed model.
3) The paper does not provide extensive validation or evaluation of the quality and realism of the event description texts generated by the LLM-based pipeline.

**Suitability:**

3

---

### Official Review · Reviewer_SWWq · 2024-05-24

**Rating:** 4
**Confidence:** 4

**Summary:**

The authors address the event traffic forecasting problem, focusing on modeling traffic patterns influenced by events. Leveraging the rapid advancements in large language models (LLMs), this paper constructs a multimodal event traffic dataset incorporating event text generated by LLMs. To tackle this problem, the authors introduce a novel model designed to handle multimodal information, achieving state-of-the-art (SOTA) performance across three different forecasting horizons.

**Strengths:**

1. The paper is well-organized and easy to follow, with clear figures and a comprehensive description of the implementation details.

2. This work introduces a new problem in the traffic forecasting domain. To my knowledge, it is the first to provide a multimodal event traffic forecasting dataset, which is highly valuable for the field. The authors reasonably use large language models to generate formatted descriptions of events.

3. The baselines presented in this paper, including PatchTST, D2STGNN, and STAEformer, are high-quality and timely, setting a high standard for future research.

**Limitations:**

1. Table 3 appears to be redundant, especially since Table 4 provides sufficient details on the results of T3.

2. Although the superior performance of the proposed T3 model is reasonable due to its integration of additional event information and raw time series representation, T3 does not show a significant performance advantage over other baselines. For example, GWNet and STAEformer exhibit similar performance. Given that SZCEC is a dataset specifically designed for event forecasting, T3 should demonstrate a more substantial performance improvement.

3. In Figure 6, the t-SNE analysis shows the event embeddings clustered into various groups. However, the authors should clarify whether all events have been plotted. The figure shows 34 dots, while there should be 32 events in SZCEC according to the authors.

4. The proposed SZCEC dataset is one of the main novelties of this work. Therefore, more analysis on it is expected. For instance, the authors could further explore the relevance among events within each cluster, given they have already illustrated the visualization of event embeddings.

5.  More recent works could be cited or selected as convincing baselines like HimNet[1],PGCN[2], and MoST[3].
[1] Dong, Zheng, et al. "Heterogeneity-Informed Meta-Parameter Learning for Spatiotemporal Time Series Forecasting." arXiv preprint arXiv:2405.10800 (2024).
[2] Shin, Yuyol, and Yoonjin Yoon. "PGCN: Progressive graph convolutional networks for spatial–temporal traffic forecasting." IEEE Transactions on Intelligent Transportation Systems (2024).
[3] Deng, Jiewen, et al. "Multi-Modality Spatio-Temporal Forecasting via Self-Supervised Learning." arXiv preprint arXiv:2405.03255 (2024).

In conclusion, I believe this paper is marginally above the acceptance standard for being the first to propose a multimodal event time series dataset. While the performance of the proposed model needs more detailed description and experimentation, the authors provide a novel method for building multimodal time series event datasets.

**Suitability:**

3

---

### Official Review · Reviewer_pDW4 · 2024-05-25

**Rating:** 4
**Confidence:** 3

**Summary:**

This paper focuses on the problem of urban traffic forecasting under large events such as sport games, exibitions, concerts, etc. Existing traffic forecasting methods cannot fully consider the influence of these events thus having suboptimal performance in these cases. This paper proposed to solve this problem by utilizing descriptional text of events as additional information to help deep learning model learn to forecast better under events. To this end, this paper developed an LLM-based event descriptional text generation pipeline, and built a dataset, known as SZCEC, for evaluation of urban traffic forecasting methods under large events. This paper proposed a multimodal event traffic forecasting model, known as T3, that considers both traffic signal and event descriptional text for event traffic forecasting. They conduct experiments on the proposed dataset and showed good performance of T3.

**Strengths:**

1. The problem studied in this paper is important which can indeed broaden the application of traffic forecasting and satisfy real world needs.
2. It is novel to utilize LLMs to generate uniform descriptional text of events and use the texts to provide event information in a multi-modal traffic forecasting.
3. The dataset built by the authors is a contribution to the related research community and can help future research of multi-modal modeling for traffic forecasting and smart cities.

**Limitations:**

1. It would be better if the related work section can discuss more on how is the event information is utilized in the existing methods, and maybe, how LLM-generated texts are utilized in other similar multi-model learning tasks.
1. The proposed SZCEC dataset only considers the events held at one exibition center in the city, but for large cities, there can be other events happends elsewhere in the city at the same time that also affect the traffic signal.
2. It is not clearly specified how the event text descriptions are generated, e.g., what LLM and what prompt are used?
3. Some designs in T3 are not well motivated, e.g., why the traffic encoder needs to be frozen, can it be trained end-to-end from scratch, or fine-tuned during the training process described in Figure 5?
4. The experiments mainly compared the proposed T3 model which uses both traffic signal and event information as input, with baselines only using the traffic signal, which is not so fair. Although the experiment in Section 6.3 compared T3 with GWNet with handcrafted event features, it is not comprehensive enough. It would be better to compare with all the baselines that also have certain event information, such as handcrafted event features, as input.
5. This paper needs further proofreading to avoid typos and grammar mistakes.

**Suitability:**

3

---

### Official Review · Reviewer_jEvq · 2024-05-27

**Rating:** 5
**Confidence:** 4

**Summary:**

This work entitled as "Event Traffic Forecasting with Sparse Multimodal Data" propose to leverage large language models (LLMs) to assist in the spatial-temporal analysis of traffic patterns. The key innovation lies in the integration of textual data with traditional traffic sensor data to predict traffic conditions during events such as sports games, concerts, and exhibitions. The authors utilize pre-trained LLMs to extract semantic information from event descriptions and fuse this information with traffic sensor data using a gated convolutional neural network. This approach addresses the challenge of sparse multimodal data and the nonlinear correlations between event-related traffic changes. The model is evaluated on a real-world dataset from Shenzhen City, demonstrating its effectiveness in capturing the impact of events on traffic patterns. The experimental results show that the proposed model outperforms baseline models.

**Strengths:**

1. Novel Integration of LLMs on STG Learning: The paper effectively integrates large language models with traffic forecasting, demonstrating the potential of using semantic information from text data to enhance spatial-temporal predictions.
2. Addressing Sparse Data Challenge: The proposed method tackles the challenge of sparse multimodal data, providing a robust solution for scenarios where traditional data sources are insufficient.
3. Real-World Application: The model is tested on real-world data from Shenzhen City, showcasing its practical applicability and effectiveness in a real-world context.
4. Performance Improvement: The experimental results highlight significant improvements in prediction accuracy compared to baseline models, validating the proposed approach.
5. Detailed Methodology: The paper provides a comprehensive description of the model architecture and the experimental setup, ensuring transparency and reproducibility.

**Limitations:**

1. Limited Dataset Diversity: The evaluation is conducted on a single dataset from Shenzhen City, which may limit the generalizability of the results to other geographical locations or types of events.
2. Complexity of Implementation: The integration of LLMs and the proposed model architecture may require substantial computational resources and expertise, which could limit its adoption in resource-constrained environments.

**Suitability:**

3

---

### Meta-Review · Area_Chair_q8jd · 2024-07-03

**Recommendation:** Accept (Poster)
**Confidence:** 4

**Metareview:**

The authors propose a multi-modal event traffic forecasting model that integrates textual and traffic data. The model was initially evaluated on a single dataset from Shenzhen City and this was a limitation of the work as highlighted by a couple of the reviewers. All the reviewers agree that the idea is very valuable, the work done is innovative and the integration of the two sources of information is well designed. Reading the rebuttal, I think the main objections of reviewer "WQWg" are dealt by the authors, in particular the issue related to the ablation of some components of the framework. For sure, the final version of the work should report the results on the two datasets, the current one and the one the authors have reported initial results in the rebuttal.